# SUBSPACE GRID-SWEEP: ML DEFENSE EVALUATION VIA CONSTRAINED BRUTE-FORCE SEARCH

## ABSTRACT

It is becoming increasingly imperative to design robust ML defenses. However, recent work has found that many defenses that initially resist state-of-the-art attacks can be broken by an adaptive adversary. Attacks can initially make defenses look strong by not finding potential adversarial examples due to obfuscated gradients, limited compute, unlucky initialization, etc. In this work, we make steps towards more reliable defense evaluation by introducing a new defense evaluation tool, Subspace Grid-sweep, that leverages deterministic inference to more simply evaluate adversarial robustness. We use Subspace Grid-sweep to show that a previously published, but now broken, defense could have been known to be broken without performing a fully adaptive attack. In order to make Subspace Grid-sweep applicable to random defenses, we show how to make deterministic variants of random defenses while retaining similar empirical effectiveness. As a result, we show that randomness may not be necessary for these defense's robustness.

## 1 INTRODUCTION

Given the increasing prevalence of machine learning models, their reliability in adversarial environments has gained much attention (Jia et al., 2020; Cohen et al., 2019; Rauber et al., 2017; 2020; Croce et al., 2021; Dong et al., 2019; Grathwohl et al., 2020; Nie et al., 2022; Madry et al., 2018). The ensuing developments towards more robust ML models led to many proposed defenses (Cohen et al., 2019; Grathwohl et al., 2020; Nie et al., 2022) and methods to evaluate these defenses (Athalye et al., 2018a; Rauber et al., 2017; Croce et al., 2021; Rauber et al., 2020).

Incorporating *randomness* into ML defenses has appeared valuable. Some of the most effective empirical defenses rely on randomness (Nie et al., 2022), and many provably robust defenses are random (Cohen et al., 2019). Randomness has been used to disrupt iterative white-box attacks (Xie et al., 2018; Raff et al., 2019) and degrade black-box attacks (Dong et al., 2019). It is no surprise then that many ML defenses have random components (Raff et al., 2019; Song et al., 2018).

Unfortunately, evaluating ML defense robustness has proven challenging. Initially promising defenses are often quickly broken (Tramèr et al., 2020; Athalye et al., 2018a). One reason for this is that defenses can seem strong because they break typical attacks (Rauber et al., 2017; Croce et al., 2021; Dong et al., 2019). However, attacks are necessarily constrained to only searching a tiny fraction of the input space given limited available compute. This search is influenced by the mechanisms, initialization, and heuristics that make up the attack, and the information (e.g., gradients) the attack can extract from the classifier. Ideally, an evaluation that could instead uniformly search the full input space (within some distance of the original input) for perturbations that would cause misclassification would be a more consistent measure of robustness. For example, an ideal evaluation of an ML defense could directly test every possible perturbation (possibly constrained by an $L_p$ norm) for several given inputs and report whether the defense successfully decreases the pervasiveness of adversarial perturbations compared to the original, undefended base classifier. Unfortunately, given high input dimensionality (e.g., 3072 for CIFAR-10), this is almost never feasible.

Moreover, a leading factor further contributing to the difficulty of evaluating robustness is randomness (Gao et al., 2022). Despite the above mentioned benefits, randomness also removes the ability to quickly check whether an underlying loss gradient is smooth, directly inhibits the use of black-box attacks for evaluation (Dong et al., 2019), and often requires an evaluator to specially craft an attack to bypass the specific flavor of randomness included (Tramèr et al., 2020).

Our work seeks to approximate that ideal evaluation and mitigate the difficulties caused by randomness while retaining the benefits of randomness in practice. We side-step the problem of searching the entire high-dimensional space of perturbations by instead only searching a lower-dimensional subspace of perturbations. We also show that, for the random defenses we study, their empirical robustness can be replicated in a deterministic manner. This allows us to bypass the obstacles to ML defense evaluation inherent to randomness and gives us new insight into published work.

Specifically we make the following contributions:

- We introduce a new brute-force method to better evaluate *deterministic* ML defenses' robustness called Subspace Grid-sweep and demonstrate its efficacy by verifying the findings of a prior, more complex, robustness evaluation (section 3).

- To use this tool on randomized defenses, we also show that several randomized defenses can be converted to be deterministic and still retain the same empirical robustness, illustrating that randomness is not necessary for empirical robustness in these defenses (section 4).

- In section 5, by combining the above two contributions, we provide new evidence some published defenses are indeed increasing robustness, while other defenses may actually be decreasing robustness, but just making those adversarial examples harder to find.

## 2 RELATED WORK

### 2.1 ML DEFENSE EVALUATION

**ML attacks.** Adversarial ML attacks create *adversarial examples*, or inputs that are minimally perturbed but cause a targeted ML model to misclassify them. To ensure that the input's ground-truth label does not change, attacks set a maximal perturbation size $\epsilon$ under some metric $L_p$, where $p \in \{0, 1, 2, \infty\}$. Researchers developed standardized suites of attacks (Rauber et al., 2017; 2020; Croce et al., 2021; Dong et al., 2019) to more fairly benchmark and compare robustness of defenses. These suites include both black-box attacks (e.g., Boundary Attack (Brendel et al., 2018)) and white-box attacks (e.g., PGD Madry et al. (2018)).

**ML defenses.** Researchers have proposed many ML defenses (Madry et al., 2018; Pang et al., 2018; Nie et al., 2022; Grathwohl et al., 2020; Cohen et al., 2019; Jia et al., 2018; 2020; Song et al., 2018; Xie et al., 2018; Raff et al., 2019). This includes adversarial training, where an ML model is exposed to adversarial examples during training, making it less susceptible to attack (Madry et al., 2018). Other defenses include detection of adversarial examples (Pang et al., 2018), input refinement (Nie et al., 2022; Grathwohl et al., 2020; Jia et al., 2018; Song et al., 2018), and other pre-processing techniques (Xie et al., 2018; Raff et al., 2019).

When a defense reports high empirical robustness, this often does not tell us *why* the defense succeeds. Prior work has shown that empirical robustness evaluations can be misleading, e.g., due to obfuscated gradients that hinder some attacks but do not reduce ML model vulnerability (Athalye et al., 2018a; Tramèr et al., 2020; Carlini & Wagner, 2017). Such defenses are vulnerable to adaptive attacks that specifically target weak components of the defense (Tramèr et al., 2020). Our work take steps towards a more useful and informative ML defense evaluation.

Randomness is necessary in certain certified robust defenses (Cohen et al., 2019; Jia et al., 2020; Levine & Feizi, 2019), but it has also been used to claim increased robustness to black-box attacks (Dong et al., 2019) and white-box attacks (Athalye et al., 2018b; Xie et al., 2018). However, in a white-box threat model, randomness in defenses can be unhelpful (Gao et al., 2022) or give a false sense of security (Athalye et al., 2018a;b). To illustrate, one prior published defense states that "Randomization makes the network much more robust to adversarial images, especially for iterative attacks (both white-box and black box)..." (Xie et al., 2018, section 1). Yet, later work broke this defense by incorporating the defense's random transformations into the attack (Athalye et al., 2018a)—a general attack strategy that is often referred to as Expectation Over Transformation (EOT) (Athalye et al., 2018b). Other randomized defenses have suffered similar fates (Gao et al., 2022). It is thus unclear when randomness in ML defenses is actually helpful, or just an unnecessary component that complicates defense evaluation.

## 2.2 PUBLISHED RANDOMIZED DEFENSES

Random defenses exhibit robustness both empirically (Nie et al., 2022; Grathwohl et al., 2020) and provably (Cohen et al., 2019; Levine & Feizi, 2019; Jia et al., 2020). In this work, we create deterministic analogs of three randomized defenses to understand if randomness is necessary for them to resist white-box attacks.

**Randomized smoothing.** Randomized Smoothing (Cohen et al., 2019) operates by taking the majority vote of a classifier over many randomly corrupted copies of the input (with the option to abstain prediction if the vote counts are too close). This method of prediction amounts to a Monte Carlo estimation of the distribution of classes around the input (within some $L_p$ ball), which can be used to prove a robustness certificate on the prediction's correctness (Cohen et al., 2019).

**Energy-based model defense.** This defense (Grathwohl et al., 2020), referred to as *EBM–Def* in this work, uses a predictor of the likelihood that the input belongs to the data distribution (i.e., an energy function). *EBM–Def* first adds random noise to the input, and then "refines" this noisy input by nudging it in the direction of increasing energy, hopefully eliminating any adversarial perturbations in the process. The defense can also aggregate multiple parallel corrupt-refine processes (referred to as "Markov chains") using different initial corruptions. As the refinement process is deterministic, the only random component of this defense is the initial noise addition.

**Diffusion models for adversarial purification.** This defense, referred to as *DiffPure*, (Nie et al., 2022) is similar to *EBM–Def* in that inputs are first corrupted (via random sampling) and then *denoised* (i.e., refined) to remove adversarial perturbations. The difference is that *DiffPure* uses a pre-trained diffusion model for the refinement, which is itself randomized (the diffusion process iteratively solves a reverse stochastic differential equation (Nie et al., 2022)). This means randomness exists in both *DiffPure*'s initial corruption and refinement processes. In contrast, *EBM–Def* refinement process is fully deterministic.

## 3 SUBSPACE GRID-SWEEP: A NEW EVALUATION TOOL

We begin with our first contribution: an evaluation tool that takes advantage of deterministic inference to provide a simple brute-force robustness evaluation.

In short, this evaluation works by exhaustively searching for regions in the input space around the original point that cause a classifier to misclassify, referred to as *adversarial regions*. By directly testing if these adversarial regions exist via a grid search (i.e., trying all the perturbations possible to a specific granularity), we can check to see if gradient-guided methods, executed in the same space, can also find any adversarial regions. If the gradient-guided methods cannot find any adversarial examples for a given input (i.e. a CIFAR-10 picture of a dog), but the grid-search did, this could be indicative of obfuscated gradients (Athalye et al., 2018a).

This evaluation measures two effects of a defense:

- Does the defense, compared to a baseline (usually the base classifier it is protecting) increase, decrease, or not affect the prevalence of successful adversarial examples?
- Does the defense hinder gradient-based attacks ability to find existing successful adversarial examples (e.g., gradient obfuscation (Athalye et al., 2018a))?

### 3.1 SEARCHING A SUBSPACE

Ideally, we would brute-force search the entire nearby input space for adversarial examples, but given high input dimensionality, this is not feasible. We side-step this limitation by instead searching a lower-dimensional (e.g., 1–6 dimensions) subspace within the nearby input space.

Subspaces are defined by $K$ orthonormal basis vectors, each of length $D$, where $D$ is the number of dimensions of the original space (e.g., 3072 for CIFAR-10, 784 for MNIST) and $K$ is the number of dimensions of the subspace. These vectors form a matrix $M$ with shape $K \times D$ which we use to project any point in the subspace to the original input space (and vice versa).

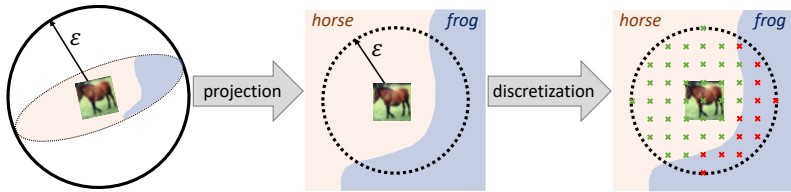

High dimensional $\ell_p$ ball     Low dimensional projection     Adversarial grid search

Figure 1: In order to search for adversarial regions nearby the original point, we take a small slice of the full high-dimensional space (i.e., a subspace). Within this subspace, we compare whether grid-sweep and gradient-guided attacks can find adversarial examples. If gradient-guided attacks cannot find any adversarial examples, but grid-sweep does, this may indicate obfuscated gradients.

Within this subspace, we execute a *grid-sweep* search. This search discretizes each axis of the subspace (where each axis is defined by a basis vector) into a preset number of bins $B$. For a subspace with $K$ dimensions, this makes the number of points we search in the subspace $= B^K$ if we use the $L_\infty$ distance, and $\leq B^K$ if we use the $L_2$ distance. Algorithm 1 provides more detail.

**Function** `GridSweepPerturbationsL2(`$K, B, M, \epsilon$`)`
    **for** $g \in \{0, ..., B\}^K$ **do**
        $g \leftarrow (g/B) * 2 - 1$ `// scale grid to [`$-1$`,1] and origin center`
        **if** $L_2\_norm(g) \leq 1$ **then**
            $\delta \leftarrow gM * \epsilon$ `// project coordinate to input space and scale`
            **yield** $\delta$ `// yield perturbation to try`
            **Algorithm 1:** Generating Grid Sweep Perturbations

Note that this grid-sweep will miss adversarial examples that do not lie on the discretized grid. To ensure this was not an issue, we also randomly selected coordinates $g$ (ignoring the equal-spacing of the grid points) within the K-dimensional subspace (and distance constraint) using the same number of samples as grid-sweep, and found little difference. These results are in App. A. Finally, we compare the results of these "blind" attacks with an execution of AutoAttack's A-PGD (Croce et al., 2021; Croce & Hein, 2021) also restricted to the continuous K-dimensional subspace.

By comparing these different methods of finding adversarial inputs, we can discover *attack bugs* where the gradient-guided attack (A-PGD in our case) fails to find adversarial examples with inputs that grid-sweep succeeded on. If such bugs are prevalent, then the defense is likely to be causing some form of gradient obfuscation (Athalye et al., 2018a).

## 3.2 CASE STUDY: SUBSPACE GRID-SWEEP FOR k-WINNERS-TAKE-ALL

We illustrate the utility of Subspace Grid-sweep for evaluating a deterministic published ML defense, k-Winners-Take-All (Xiao et al., 2020), that is known to suffer from (highly) obfuscated gradients (Tramèr et al., 2020).

We search for adversarial examples within small subspaces consisting of 1–6 dimensions using grid-sweep and A-PGD and fix the maximum adversarial $L_2$ distance to $\epsilon = 0.5$. For A-PGD, we use AutoAttack's implementation (Croce et al., 2021; Croce & Hein, 2021) with 100 steps and 20 restarts. Experiments are completed on both the CIFAR-10 and SVHN validation datasets, the same datasets used by

**Input** : dataset
**Output:** num_vuln, apgd_found
$n \leftarrow \text{len(dataset)}$;
num_vuln $\leftarrow 0$;
apgd_found $\leftarrow 0$;
**for** $x$ *in dataset* **do**
    **if** *gridsweep_finds_adv_example*$(x)$ **then**
        num_vuln $\leftarrow v + 1$;
        **if** *apgd_finds_adv_example*$(x)$ **then**
            apgd_found $\leftarrow$ apgd_found $+ 1$;
robust_against_gridsweep $\leftarrow 1 - \left(\frac{\text{num\_vuln}}{n}\right)$;
**Algorithm 2:** Subspace Grid-sweep

the original work. For grid-sweep, the number of discretized search bins $B$ varies between 9 and 1001, using fewer bins as the subspace dimensionality increases. More detailed results and values

Table 1: Subspace Grid-sweep results for k-Winners-Take-All showing increased adversarial example prevalence and signs of gradient obfuscation. Read section 3.2 for full reasoning. First column shows robustness against Grid-sweep, as defined in algorithm 2. Second and third columns show the number of vulnerable datapoints grid-sweep found how many of those A-PGD found (where vulnerable means at least one successful adversarial example was found).

| Model | Robustness against Grid-sweep | # Vuln Datapoints Found | | Model | Robustness against Grid-sweep | # Vuln Datapoints Found | |
|---|---|---|---|---|---|---|---|
| | | Grid-sweep | A-PGD | | | Grid-sweep | A-PGD |
| Undefended | **0.95** | 144 | **144**/144 | Undefended | **0.98** | 74 | **74**/74 |
| kWTA-defended | 0.61 | 1160 | **1011**/1160 | kWTA-defended | 0.71 | 896 | **787**/896 |
| (a) CIFAR-10 | | | | (b) SVHN | | | |

of $B$ and $K$ for these experiments are in Tables 4–5 in App. A. Aggregate measures summarizing experiments over all subspaces are used in this section for brevity.

We calculate and present the results following algorithm 2 to better show the two effects previously described: (1) whether a defense changes the prevalence of adversarial examples; and (2) whether a defense is hindering gradient-guided attacks from finding existing adversarial examples.

The results are summarized in in Table 1. As mentioned in section 3.1, there are two effects of the defense we should observe.

1. Comparing the *robustness against grid-sweep* between the undefended model[1] with the k-Winners-Take-All-defended model, we find that the robustness against grid-sweep (defined in algorithm 2) is lower for the k-Winners-Take-All-defended model. This indicates that this defense is *increasing* the prevalence of adversarial examples.

2. Comparing the number of vulnerable datapoints between grid-sweep and A-PGD for the k-Winners-Take-All-defended model, we also notice that A-PGD often fails to find adversarial examples against k-Winners-Take-All in cases where grid-sweep would find them. This shows that k-Winners-Take-All is making it harder for gradient-based methods to find adversarial examples, even though they still exist. Note that A-PGD on the undefended model finds all vulnerable datapoints as grid-sweep, indicating no hindrance.

## 4 DETERMINISTIC DEFENSES ARE SIMILARLY (WHITE-BOX) ROBUST

An important feature of Subspace Grid-sweep is that its search covers the entire subspace (within some distance from an input) to a certain granularity, ensuring that existing adversarial regions (not small enough to fit between grid points) will be discovered. This feature assumes that adversarial regions are static and non-probabilistic. As adversarial regions are an abstraction of classifier/ML defense behavior, this means Subspace Grid-sweep only works on deterministic ML defenses.

In this section, we investigate whether we can remove randomness [2] and yet retain the claimed empirical benefits of some randomized defenses. To do this, we begin by constructing deterministic variants of each of the three randomized defenses introduced earlier (Randomized Smoothing (Cohen et al., 2019) (section 4.1), Energy-based Models as a Defense (Grathwohl et al., 2020) (section 4.2), and Diffusion Models for Adversarial Purification (Nie et al., 2022) (section 4.3)). We then compare the robustness of the random and deterministic variants of these defenses against a white-box attack.

For consistency, we evaluate both variants of the defenses (randomized and deterministic) using each defense's published code and the implementation of PGD that the original work used for evaluation. Specifically, for Randomized Smoothing, we implemented PGD as described in the original paper (Cohen et al., 2019). For *EBM–Def*, we used foolbox (Rauber et al., 2017; 2020) to execute PGD with a binary-search. Finally, we executed A-PGD from AutoAttack (Croce et al., 2021)

---

[1]This is the baseline model from the original paper, but without k-Winners-Take-All.

[2]'Removing randomness' consists of modifying the defense to ensure that a given input will always have the same output. This property is sufficient to avoid issues caused by randomness discussed in section 6.

against *DiffPure*. All attacks were executed, as in the original works, by using test datapoints from the CIFAR-10 dataset (Krizhevsky, 2009) as inputs.

Overall, we demonstrate that the deterministic analogs have similar or equal empirical robustness to the original random defenses, showing that the robustness claimed in these works can also be achieved via a deterministic mechanism, and that randomness is not necessary.

## 4.1 RANDOMIZED SMOOTHING

As described in section 2.2, Randomized Smoothing aggregates the inferences of several Gaussian-corrupted copies of the input through the *base classifier* (the classifier being defended).

**Making this defense deterministic.** To make this process deterministic, we fix the random seed before every prediction and use the same set of corruptions for every inference. We refer to this deterministic version of inference as *Deterministic Smoothing*.

**Comparing the randomized and deterministic versions.** To compare the robustness of these defenses, we execute Projected Gradient Descent (PGD) (Madry et al., 2018) on both Randomized Smoothing and Deterministic Smoothing using 100 steps, a step size of 0.025 with an $L_2$ distance constraint of 0.5. The base classifier used is provided by the original work (Cohen et al., 2019), and has been smoothed with $\mathcal{N}(0, 0.5)$. We execute a step towards the increasing loss gradient aggregated across $n$ noise-corrupted copies of the original image. For Deterministic Smoothing, these

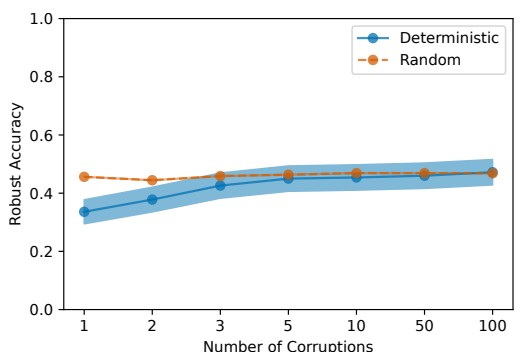

Figure 2: PGD success against randomized and deterministic smoothing converges to the same empirical robustness as number of corruptions increases. Shaded regions are 95% confidence intervals (Random's confidence interval is very small).

noise-corrupted copies do not change across different PGD steps, resulting in a less noisy gradient. However, for Randomized Smoothing, the sampled noise used to corrupt $n$ copies of the input is re-sampled and 100 EoT (Expectation over Transformation (Athalye et al., 2018b)) gradient estimates are conducted for each step. To ensure our results were not affected by PGD non-convergence, we also executed PGD for up to 10000 steps for 100 corruptions, and found the same results. In section 5, Deterministic Smoothing also exhibits similar robustness against A-PGD.

Figure 2 shows the robust accuracy of the randomized and deterministic variants as we increase the number of noise-corrupted copies $n$ used to make predictions. The robust accuracy of both Randomized Smoothing and Deterministic Smoothing rapidly converge to the same value at around 5 corruptions, which is many fewer than the 100 corruptions used in the original work (Cohen et al., 2019). This indicates that the empirical robustness of Randomized Smoothing does not come from the randomness of the noise, but from the *self-ensembling* effect of aggregating multiple inferences within proximity of the original point.

## 4.2 ENERGY-BASED MODELS

Recall from section 2.2 that the only source of randomness in *EBM–Def* is the random corruption added before the deterministic refinement stage using the trained energy-based model (EBM).

**Making this defense deterministic.** Similar to Randomized Smoothing, we make the initial added corruptions the same for each inference by setting the same random seed before each prediction.

**Comparing the randomized and deterministic versions.** The *EBM–Def* paper uses fool-box (Rauber et al., 2017; 2020) to execute PGD with a binary-search to create adversarial examples from the CIFAR-10 dataset (Krizhevsky, 2009) with a minimal $L_2$ or $L_\infty$ distance to the original point. PGD is executed using 20 random restarts, with 40 steps targeting both variants of *EBM–Def*.

Using this type of attack, a successful adversarial example is found for every test datapoint, and the distance to the original point is used to measure the robustness of the targeted classifier, with larger distances corresponding to more robustness.

The deterministic and randomized versions of *EBM–Def* behave nearly identically. Figure 3 shows their robustness using 1, 2 or 5 "markov chains" (i.e., parallel noise-refine processes on copies of the input), for four different magnitudes for the adversarial perturbations. Thus, the empirical robustness gained by *EBM–Def* appears to be due to the deterministic refinement, rather than the randomness of the initial corruptions. For more detailed plots that mirror the original work, see figure 5 in App. B.

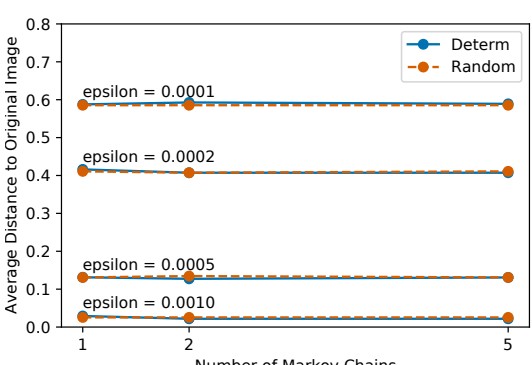

Figure 3: Attacking randomized and deterministic versions of *EBM–Def* yields near-identical empirical robustness, at four different magnitudes for the adversarial perturbations.

### 4.3 DIFFUSION
MODELS FOR ADVERSARIAL PURIFICATION

Diffusion Models for Adversarial Purification (Nie et al., 2022) also adds random noise to the input, but then applies a denoising stage that is itself randomized.

**Making this defense deterministic** As for Deterministic Smoothing and *EBM–Def*, we always use a fixed seed for sampling the initial input noise. For *DiffPure*, we further use the same seed for the denoising process, thereby making it deterministic as well. To smoothly interpolate between this fully deterministic version and the original randomized version, we also consider a defense variant that introduces increasingly more variability by cycling through $k$ different seeds in subsequent inferences, where $k \in \{1, 2, 3, 5, 10, 50, 100\}$.

**Comparing the randomized and deterministic versions** The original *DiffPure* work measures robustness by executing AutoAttack (Croce et al., 2021) with test samples from the CIFAR-10 dataset using 20 EoT gradient estimates with 1 random restart. We mirror this evaluation by executing AutoAttack in both its random and standard settings, with $L_2$ and $L_\infty$ constraint distances of 0.5 and 0.031373, respectively. We use A-PGD to compare versions of *DiffPure*, as it is the only attack executed in all of these settings.

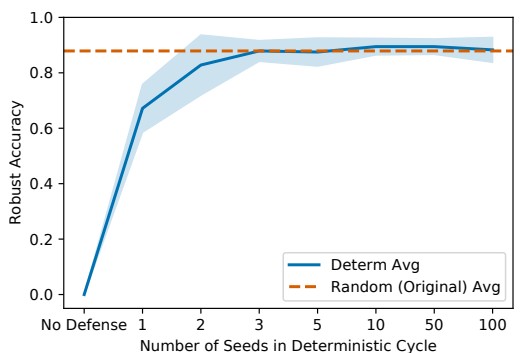

Figure 4: Deterministic analogs of *DiffPure* are just as (or nearly as) robust as the random original.

Figure 4 shows that *DiffPure* and its deterministic analog rapidly converge in empirical robustness as we increase the number of seeds used to sample random noise in subsequent iterations. The completely deterministic version of *DiffPure* (1 seed) exhibits 76% of the empirical robustness of the original randomized version (67% out of 88%). With a cycle of just three random seeds, we recover the empirical robustness of the original defenses. Thus, this defense's empirical robustness appears to stem primarily from the, now deterministic, diffusion denoising process.

## 5 SUBSPACE GRID-SWEEP ON DETERMINISTIC VARIANTS OF RANDOM DEFENSES

Section 3 confirmed prior work's finding that k-Winners-Take-All is flawed (Tramèr et al., 2020). In this section, we use Subspace Grid-sweep on the deterministic variants of the previously-published

random defenses described in section 4 to evaluate their robustness. Overall, our results show that Randomized Smoothing successfully reduces the prevalence of adversarial examples, while *EBM–Def* and *DiffPure* may actually be *increasing* the prevalence of adversarial examples, and only making it harder for gradient-based attacks to find them.

Each defense we consider uses a different underlying base classifier. To create a fair comparison, we first execute Subspace Grid-sweep against each defense's undefended base classifier (referred to as *Base* in Table 2), and then compare it to the deterministic variants of the defenses from section 4. Because *DiffPure*'s inference is computationally expensive (which makes running A-PGD prohibitive), we created a faster version—referred to as *DiffSmall*—that uses fewer refinement steps. We anticipated it would not be as robust as the published *Diff-Pure* (Nie et al., 2022), but we expected similar behavior in Subspace Grid-sweep. In App. A, we provide detailed tables of our full results including measures for statistical significance.

Table 2: First column shows robustness against Grid-sweep, as defined in algorithm 2. Second and third columns show the number of vulnerable datapoints grid-sweep found and how many of those A-PGD found. Read section 5 for analysis.

| Model | Robustness against Grid-sweep | # Vuln Datapoints Found | |
|---|---|---|---|
| | | Grid-sweep | A-PGD |
| (Smoothing) Base | 0.97 | 74 | **73**/74 |
| (Smoothing) Defended | **0.98** | 70 | **27**/70 |
| (*EBM–Def*) Base | **0.99** | 34 | **34**/34 |
| (*EBM–Def*) Defended | 0.96 | 104 | **92**/104 |
| (*DiffSmall*) Base | **0.95** | 144 | **144**/144 |
| (*DiffSmall*) Defended | 0.48 | 1572 | **867**/1572 |

**Deterministic Smoothing.** The first two rows of Table 2 show that the base (undefended) classifier has a robustness against grid-sweep of 97% and the Deterministic Smoothing-defended classifier is slightly more robust at 98%. Also, A-PGD successfully finds nearly all vulnerable points on the base classifier, but misses many when it is defended by Deterministic Smoothing. These results indicate two insights: 1) Deterministic Smoothing successfully reduces the prevalence of adversarial examples compared to the undefended classifier and; 2) A-PGD (from AutoAttack (Croce et al., 2021)) sometimes misses the, albeit smaller, adversarial regions that still exist. We hypothesize this is because of a slight misalignment between the objective A-PGD tries to maximize and the inference procedure of Deterministic Smoothing. Specifically, A-PGD uses the mean of the loss gradients across all (deterministic) corruptions as prescribed by EoT (Athalye et al., 2018b), incentivizing a perturbation that increases the loss for all corruptions. However, to fool Deterministic Smoothing, we simply need to fool a plurality of corruptions. An attack that first finds a plurality of *easy* corruptions to fool, then only focuses on those may be more appropriate. However, we find the standard A-PGD implementation to be sufficient for our purposes.

***EBM–Def*.** For this defense, we find that the underlying base classifier is already very robust against grid-sweep at 99%. Comparing with the *EBM–Def*-defended classifier yields two insights: 1) ironically, adding *EBM–Def* increases the prevalence of adversarial examples in the low-dimensional subspaces searched; and 2) *EBM–Def* may have a slight gradient obfuscation issue, as A-PGD fails to find an adversarial example in around 10% of inputs that Grid-sweep does find one.

***DiffSmall*.** For *DiffSmall*, we find that the defense strongly increases the prevalence of adversarial examples in the subspaces searched compared to the undefended classifier. Indeed, robustness to grid-sweep is reduced from 95% to 48% when the diffusion defense is in place compared to the undefended base classifier. More worryingly, this decrease in robustness is partially missed by A-PGD. Indeed, the gradient-based attack fails to find many of the vulnerable datapoints that were found with with either a grid-sweep or random sampling approach.

## 6 DISCUSSION

In this section, we remark on randomness in evaluation, some limitations and theoretical aspects of this work, and promising future work.

**Problems with randomness in evaluation.** As pointed out in section 2.1, confidently figuring out whether an ML defense can be defeated is difficult. Including randomness adds more difficulty.

For example, randomness, sometimes by design (Xiao et al., 2020), is sufficient to remove the ability to quickly check whether an underlying loss gradient is smooth. While this difficult loss surface could be the result of a robust defense, it also could be obfuscating a weak defense that can be defeated via an adaptive attack (Tramèr et al., 2020; Gao et al., 2022). Additionally, while black-box attacks can be useful to expose gradient masking (Athalye et al., 2018a), randomness directly degrades the ability to use this class of attack for evaluation (Dong et al., 2019), requiring potential evaluators to forgo them as tools. Similarly, creating attacks that can expose the underlying vulnerability of randomized defenses often requires an evaluator to specially craft an attack to bypass any specific flavor of randomness included (Tramèr et al., 2020).

As shown in section 4, the extra effort needed to expose a randomized defense's weakness may not be necessary if these defenses can be made deterministic and still benefit from empirical robustness. For this reason, random defense authors may consider providing a deterministic analog for evaluation.

**Limitations and theoretical aspects.** While the three defenses we study could be made deterministic while retaining their adversarial robustness, there exist some ML defense benefits where randomness is not easily removed or is a truly necessary component. In the cases where these benefits must be retained, our advice and new evaluation method may not be applicable.

For example, the robustness certificate gained in Randomized Smoothing is dependent on the Monte Carlo estimation of the distribution of classes within an $L_p$ ball of the original input point. The defense estimates this by adding Gaussian-sampled corruptions to the input point and aggregating the classifications into a distribution. However, in the version of this defense where the random seed is fixed (and known to an attacker) resulting in the same corruptions for each inference (i.e., Deterministic Smoothing in section 4.1), a robustness certificate can no longer be justified. The loss of the robustness certificate occurs because the deterministic corruptions are no longer a valid Monte Carlo estimation. In this case, an attacker no longer theoretically needs to attempt to optimize their adversarial perturbations against unknown Gaussian-sampled corruptions, but only against the known set of corruptions used in every inference.

For this reason, in principle, Deterministic Smoothing should be an easier defense to attack. It is interesting, then, that the empirical success of an attacker that knows the deterministic corruptions used for prediction is essentially the same as an attacker that does not know the corruptions used as shown in section 4.1 (given at least 5 corruptions are used for prediction as shown in figure 2).

**Future work.** Given this cursory discord between theoretical and empirical robustness, it could be valuable to investigate how an attacker's knowledge of a fixed random seed affects their capabilities.

Also, because randomness has been shown to be effective against black-box attacks (Dong et al., 2019), it could be valuable to see if they become more helpful for evaluating deterministic analogs of randomized attacks. It may be the case that if a black-box attack becomes more effective against a deterministic analog of a promising randomized defense, then this may be a sign of a potentially randomness-obscured weakness in the defense.

# 7 CONCLUSION

Overall, this work suggests that Subspace Grid-sweep can be a valuable tool to understand: (1) if an ML defense is likely increasing, decreasing, or not affecting the prevalence of adversarial examples and; (2) if an ML defense is hindering gradient-guided methods such as A-PGD from finding existing adversarial examples. Our findings also suggest an additional step of consideration when designing random ML defenses to check whether randomness is truly necessary. Indeed, we have shown that: (1) some ML defenses can have their randomness removed and still retain their robustness and; (2) without randomness, ML defenses can be evaluated with Subspace Grid-sweep.

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

# A    SUBSPACE GRID-SWEEP DETAILS

The included tables in this section give the robustness of all undefended and defended classifiers to all search methods in several different sub-spaces. These results were aggregated in the main paper.

To measure statistical significance, we use McNemar's test on whether Grid-sweep and A-PGD found the same set of vulnerable datapoints. P-values are given throughout the tables. We consider a p-value of less than 0.05 to be significant.

Table 3: Extended summary Subspace Grid-sweep results for k-Winners-Take-All. First column is the robustness of models when attacked by Grid-sweep. The following columns show how many vulnerable datapoints are found by each method. In contrast to the undefended classifier, A-PGD fails to find vulnerable datapoints that Grid-sweep finds in the k-Winners-Take-All-defended classifier, indicating that this defense makes them harder to find for gradient-based methods. Also, as reflected in the decreased robustness against Grid-sweep, the k-Winners-Take-All-defended classifier actually increases the prevalence of adversarial examples.

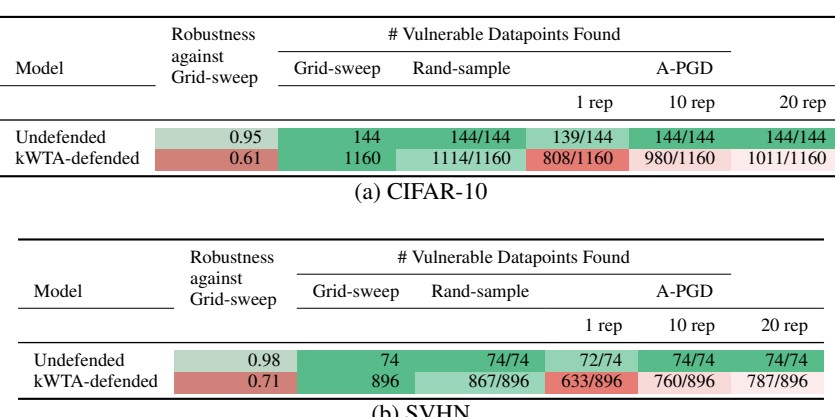

| Model | Robustness against Grid-sweep | # Vulnerable Datapoints Found | | | | |
|---|---|---|---|---|---|---|
| | | Grid-sweep | Rand-sample | A-PGD | | |
| | | | | 1 rep | 10 rep | 20 rep |
| Undefended | 0.95 | 144 | 144/144 | 139/144 | 144/144 | 144/144 |
| kWTA-defended | 0.61 | 1160 | 1114/1160 | 808/1160 | 980/1160 | 1011/1160 |

(a) CIFAR-10

| Model | Robustness against Grid-sweep | # Vulnerable Datapoints Found | | | | |
|---|---|---|---|---|---|---|
| | | Grid-sweep | Rand-sample | A-PGD | | |
| | | | | 1 rep | 10 rep | 20 rep |
| Undefended | 0.98 | 74 | 74/74 | 72/74 | 74/74 | 74/74 |
| kWTA-defended | 0.71 | 896 | 867/896 | 633/896 | 760/896 | 787/896 |

(b) SVHN

Table 4: Extended results for k-Winners-Take-All defense using Subspace Grid-sweep for CIFAR-10. For the undefended model, A-PGD finds adversarial regions (within $\epsilon$ distance of original point) for every datapoint Grid-sweep finds adversarial regions, indicating that gradient-based methods are not hindered from finding existing adversarial regions. However, A-PGD fails to find vulnerable datapoints that Grid-sweep finds in the k-Winners-Take-All-defended classifier, indicating that this defense makes them harder to find for gradient-based methods. Also, as reflected in the decreased robustness against all search methods, the k-Winners-Take-All-defended classifier increases the prevalence of adversarial examples.

| Sub-space / Grid | | | | A-PGD | | | Stat Test: A-PGD same as Grid-sweep? | | |
|---|---|---|---|---|---|---|---|---|---|
| Dims | Bins | Grid-sweep | Rand-sample | 1 rep | 10 rep | 20 rep | 1 rep p-value | 10 rep p-value | 20-rep p-value |
| Undefended | | | | | | | | | |
| 1 | 1001 | 0.97 | 0.97 | 0.97 | 0.97 | 0.97 | 0.1573 | No Diff | No Diff |
| 2 | 51 | 0.96 | 0.95 | 0.96 | 0.95 | 0.95 | 1 | 0.31731 | 0.31731 |
| 3 | 21 | 0.96 | 0.96 | 0.96 | 0.96 | 0.96 | 0.1573 | No Diff | No Diff |
| 4 | 11 | 0.95 | 0.95 | 0.95 | 0.95 | 0.95 | No Diff | No Diff | No Diff |
| 5 | 9 | 0.94 | 0.94 | 0.94 | 0.94 | 0.94 | 0.31731 | 0.31731 | 0.31731 |
| 6 | 9 | 0.94 | 0.93 | 0.93 | 0.93 | 0.93 | 0.31731 | 0.31731 | 0.31731 |
| Defended | | | | | | | | | |
| 1 | 1001 | 0.67 | 0.7 | 0.85 | 0.82 | 0.82 | 0 | 0 | 0 |
| 2 | 51 | 0.64 | 0.63 | 0.71 | 0.65 | 0.64 | 0 | 0.23932 | 0.84148 |
| 3 | 21 | 0.61 | 0.6 | 0.71 | 0.64 | 0.61 | 0 | 0.00671 | 0.65472 |
| 4 | 11 | 0.62 | 0.61 | 0.71 | 0.62 | 0.61 | 0 | 0.82726 | 0.29715 |
| 5 | 9 | 0.59 | 0.57 | 0.7 | 0.62 | 0.61 | 0 | 0.00176 | 0.07364 |
| 6 | 9 | 0.55 | 0.52 | 0.69 | 0.63 | 0.6 | 0 | 0 | 0 |

Table 5: Extended results for k-Winners-Take-All defense using Subspace Grid-sweep for SVHN. For the undefended model, A-PGD finds adversarial regions (within $\epsilon$ distance of original point) for every datapoint Grid-sweep finds adversarial regions, indicating that gradient-based methods are not hindered from finding existing adversarial regions. However, A-PGD fails to find vulnerable datapoints that Grid-sweep finds in the k-Winners-Take-All-defended classifier, indicating that this defense makes them harder to find for gradient-based methods. Also, as reflected in the decreased robustness against all search methods, the k-Winners-Take-All-defended classifier increases the prevalence of adversarial examples.

| Sub-space / Grid | | | | A-PGD | | | Stat Test: A-PGD same as Grid-sweep? | | |
|---|---|---|---|---|---|---|---|---|---|
| Dims | Bins | Grid-sweep | Rand-sample | 1 rep | 10 rep | 20 rep | 1 rep p-value | 10 rep p-value | 20-rep p-value |
| Undefended | | | | | | | | | |
| 1 | 1001 | 0.99 | 0.99 | 0.99 | 0.99 | 0.99 | 0.1573 | No Diff | No Diff |
| 2 | 51 | 0.98 | 0.98 | 0.98 | 0.98 | 0.98 | 1 | 0.31731 | 0.31731 |
| 3 | 21 | 0.98 | 0.98 | 0.98 | 0.98 | 0.98 | 0.1573 | No Diff | No Diff |
| 4 | 11 | 0.97 | 0.97 | 0.97 | 0.97 | 0.97 | No Diff | No Diff | No Diff |
| 5 | 9 | 0.97 | 0.97 | 0.97 | 0.97 | 0.97 | 0.31731 | 0.31731 | 0.31731 |
| 6 | 9 | 0.96 | 0.96 | 0.96 | 0.96 | 0.96 | 0.31731 | 0.31731 | 0.31731 |
| Defended | | | | | | | | | |
| 1 | 1001 | 0.75 | 0.76 | 0.89 | 0.86 | 0.86 | 0 | 0 | 0 |
| 2 | 51 | 0.72 | 0.72 | 0.79 | 0.75 | 0.73 | 0 | 0.23932 | 0.84148 |
| 3 | 21 | 0.71 | 0.7 | 0.78 | 0.73 | 0.72 | 0 | 0.00671 | 0.65472 |
| 4 | 11 | 0.71 | 0.71 | 0.77 | 0.73 | 0.72 | 0 | 0.82726 | 0.29715 |
| 5 | 9 | 0.71 | 0.69 | 0.78 | 0.72 | 0.71 | 0 | 0.00176 | 0.07364 |
| 6 | 9 | 0.68 | 0.68 | 0.77 | 0.71 | 0.7 | 0 | 0 | 0 |

Table 6: Extended results for Deterministic Smoothing defense using Subspace Grid-sweep. For the undefended model, A-PGD (10 repeats and 20 repeats) finds adversarial regions (within $\epsilon$ distance of original point) for every datapoint Grid-sweep finds adversarial regions, indicating that gradient-based methods are not hindered from finding existing adversarial regions. Deterministic Smoothing reduces prevalence of adversarial regions and McNemar's test finds no signs of gradient obfuscation.

| Sub-space / Grid | | | | A-PGD | | | Stat Test: A-PGD same as Grid-sweep? | | |
|---|---|---|---|---|---|---|---|---|---|
| Dims | Bins | Grid-sweep | Rand-sample | 1 rep | 10 rep | 20 rep | 1 rep p-value | 10 rep p-value | 20-rep p-value |
| Undefended | | | | | | | | | |
| 1 | 1001 | 0.93 | 0.93 | 0.95 | 0.93 | 0.93 | 0.0027 | No Diff | No Diff |
| 2 | 51 | 0.91 | 0.91 | 0.92 | 0.91 | 0.91 | 0.01431 | No Diff | No Diff |
| 3 | 21 | 0.89 | 0.89 | 0.91 | 0.89 | 0.89 | 0.00091 | 0.31731 | 0.31731 |
| 4 | 11 | 0.87 | 0.87 | 0.89 | 0.87 | 0.87 | 0.0027 | 0.31731 | 0.31731 |
| 5 | 9 | 0.86 | 0.86 | 0.88 | 0.85 | 0.85 | 0.01255 | 0.31731 | 0.08326 |
| 6 | 9 | 0.85 | 0.86 | 0.87 | 0.85 | 0.84 | 0.02092 | 0.08326 | 0.0455 |
| Defended | | | | | | | | | |
| 1 | 1001 | 0.99 | 0.99 | 0.99 | 0.99 | 0.99 | 0.08326 | 0.08326 | 0.08326 |
| 2 | 51 | 0.98 | 0.98 | 0.99 | 0.99 | 0.99 | 0.01431 | 0.01431 | 0.01431 |
| 3 | 21 | 0.98 | 0.98 | 0.99 | 0.99 | 0.99 | 0.00468 | 0.00468 | 0.00815 |
| 4 | 11 | 0.98 | 0.98 | 0.99 | 0.99 | 0.99 | 0.00815 | 0.00815 | 0.00815 |
| 5 | 9 | 0.97 | 0.97 | 0.99 | 0.99 | 0.99 | 0.00053 | 0.00091 | 0.00091 |
| 6 | 9 | 0.97 | 0.96 | 0.98 | 0.98 | 0.98 | 0.01141 | 0.01141 | 0.01141 |

# B  *EBM–Def* DETAILS

# C  COMPUTE USAGE AND INFRASTRUCTURE

To complete our experiments, we used a Linux server with 256 GB of memory, an AMD Ryzen Threadripper PRO 3975WX CPU, and a Nvidia GeForce RTX 3090 GPU. The experiments executed took two months of compute time, albeit while others were also running experiments on the server.

# D  PROBLEMS WITH RANDOMNESS IN DEPLOYMENT

It is easy to see that a non-deterministic classifier (perhaps made non-deterministic by a defense) is always more vulnerable than a deterministic classifier against a persistent and repetitive attacker. The capability to attempt multiple identical queries mirrors that of an attacker that queries a model

Table 7: Extended results for *EBM–Def* defense using Subspace Grid-sweep. For the undefended model, A-PGD finds adversarial regions (within $\epsilon$ distance of original point) for nearly every data-point Grid-sweep finds adversarial regions, indicating that gradient-based methods are not hindered from finding existing adversarial regions. However, *EBM–Def* seems to slightly increase the prevalence of adversarial examples but has few signs of gradient obfuscation.

| Sub-space / Grid | | | | A-PGD | | | Stat Test: A-PGD same as Grid-sweep? | | |
| --- | --- | --- | --- | --- | --- | --- | --- | --- | --- |
| Dims | Bins | Grid-sweep | Rand-sample | 1 rep | 10 rep | 20 rep | 1 rep p-value | 10 rep p-value | 20-rep p-value |
| Undefended | | | | | | | | | |
| 1 | 1001 | 0.99 | 0.99 | 1 | 0.99 | 0.99 | 0.1573 | No Diff | No Diff |
| 2 | 51 | 0.99 | 0.99 | 0.99 | 0.99 | 0.99 | No Diff | No Diff | No Diff |
| 3 | 21 | 0.99 | 0.99 | 0.99 | 0.99 | 0.99 | No Diff | No Diff | No Diff |
| 4 | 11 | 0.99 | 0.99 | 0.99 | 0.99 | 0.99 | No Diff | No Diff | No Diff |
| 5 | 9 | 0.98 | 0.99 | 0.98 | 0.98 | 0.98 | No Diff | 0.31731 | 0.31731 |
| 6 | 9 | 0.99 | 0.99 | 0.98 | 0.98 | 0.98 | 0.31731 | 0.31731 | 0.31731 |
| Defended | | | | | | | | | |
| 1 | 1001 | 0.97 | 0.97 | 0.99 | 0.98 | 0.98 | 0.00157 | 0.31731 | 0.5637 |
| 2 | 51 | 0.97 | 0.97 | 0.99 | 0.97 | 0.97 | 0.00468 | 0.5637 | 0.5637 |
| 3 | 21 | 0.97 | 0.97 | 0.99 | 0.97 | 0.97 | 0.0027 | 0.1573 | 0.1573 |
| 4 | 11 | 0.96 | 0.97 | 0.99 | 0.97 | 0.97 | 0.00091 | 0.1573 | 0.1573 |
| 5 | 9 | 0.96 | 0.97 | 0.99 | 0.97 | 0.96 | 0.00053 | 0.08326 | 0.31731 |
| 6 | 9 | 0.95 | 0.97 | 0.98 | 0.96 | 0.96 | 0.00031 | 0.0455 | 0.0455 |

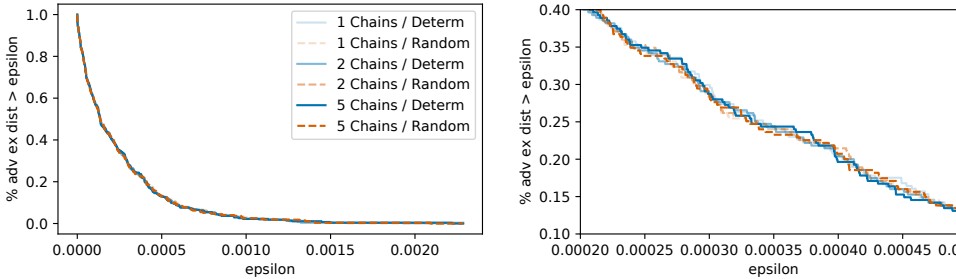

(a) The distributions of adversarial example distances found when attacking random (dashed) and deterministic (solid) *EBM–Def* using 1, 2, or 5 Markov Chains is nearly identical.

(b) Zoomed in comparison of *EBM–Def* adversarial example distances showing the slightest difference between random (dashed) and deterministic (solid) *EBM–Def* when 1 Markov Chain is used.

Figure 5: Distribution of adversarial example distances from the original point when attacking a random and deterministic *EBM–Def*.

many times during an attack to optimize their adversarial input. However, in this case, we assume the attacker only attempts the same input multiple times, with no need to optimize. Because there is no optimization, the attacker only needs to know the black-box hard-label classification for each query. The issue is visually presented in figure 6.

For a randomly-seeded classifier $f_\theta$, let $p_{(x,y)} = \Pr_{r \sim R}[f_\theta(x; r) = y]$ be the probability (over the randomness $r$) of a correct classification for input $(x, y)$. Also, denote the accuracy of a deterministic classifier by $q_{(x,y,d)} = \mathbf{1}[f_\theta(x; d) = y]$, where $d$ is a fixed random seed. To begin, note that $p_{(x,y)} = \mathbb{E}_{d \sim R}[q_{(x,y,d)}]$ on any one random sample because the fixed randomness $d$, while it does not change, is still randomly selected once. But now note that if we sample from the random classifier $N$ times, choosing a fresh random seed each time, then the probability that we correctly label the example $x$ every time is $p_{(x,y)}^N$ by independence. But for the deterministic classifier the probability it returns the correct answer remains unchanged, i.e., $\mathbb{E}_{d \sim R}[q_{(x,y,d)}]$. Therefore, we are guaranteed that $p_{(x,y)}^N \leq \mathbb{E}_{d \sim R}[q_{(x,y,d)}]$, with the inequality being a strict inequality any time $p_{(x,y)} < 1$.

Overall, including randomness in an ML classifier (i.e., by using a random ML defense) directly reduces its adversarial robustness to either a repetitive attacker or an environment where the same query can be expected many times (e.g., many host-based antivirus scanners scanning the same

Table 8: Extended results for *DiffSmall* defense using Subspace Grid-sweep. For the undefended model, A-PGD finds adversarial regions (within $\epsilon$ distance of original point) for nearly every data-point Grid-sweep finds adversarial regions, indicating that gradient-based methods are not hindered from finding existing adversarial regions. However, *DiffSmall* both increases the prevalence of adversarial examples and shows signs of gradient obfuscation.

| Sub-space / Grid | | | | | A-PGD | | | Stat Test: A-PGD same as Grid-sweep? | | |
| --- | --- | --- | --- | --- | --- | --- | --- | --- | --- | --- |
| Dims | Bins | Diffusion Steps | Grid-sweep | Rand-sample | 1 rep | 10 rep | 20 rep | 1 rep p-value | 10 rep p-value | 20-rep p-value |
| | **Undefended** | | | | | | | | | |
| | 1 | 1001 | 0.97 | 0.97 | 0.97 | 0.97 | 0.97 | 0.31731 | No Diff | No Diff |
| | 2 | 51 | 0.96 | 0.95 | 0.96 | 0.95 | 0.95 | 0.31731 | 0.31731 | 0.31731 |
| | 3 | 21 | 0.96 | 0.96 | 0.96 | 0.96 | 0.96 | 0.1573 | No Diff | No Diff |
| | 4 | 11 | 0.95 | 0.95 | 0.95 | 0.95 | 0.95 | No Diff | No Diff | No Diff |
| | 5 | 9 | 0.94 | 0.94 | 0.95 | 0.94 | 0.94 | 0.1573 | 0.31731 | 0.31731 |
| | 6 | 9 | 0.94 | 0.93 | 0.94 | 0.93 | 0.93 | 0.31731 | 0.31731 | 0.31731 |
| | **Defended** | | | | | | | | | |
| 1 | 1001 | 1 | 0.6 | 0.61 | 0.91 | 0.76 | 0.75 | 0 | 0.00006 | 0.00011 |
| 1 | 1001 | 2 | 0.55 | 0.56 | 0.93 | 0.74 | 0.71 | 0 | 0.00001 | 0.00006 |
| 1 | 1001 | 3 | 0.53 | 0.55 | 0.86 | 0.73 | 0.71 | 0 | 0.00001 | 0.00002 |
| 1 | 1001 | 5 | 0.5 | 0.52 | 0.84 | 0.72 | 0.66 | 0 | 0 | 0.00016 |
| 1 | 1001 | 10 | 0.48 | 0.46 | 0.86 | 0.7 | 0.68 | 0 | 0 | 0.00001 |
| 2 | 51 | 1 | 0.58 | 0.54 | 0.89 | 0.76 | 0.73 | 0 | 0.00002 | 0.00011 |
| 2 | 51 | 2 | 0.52 | 0.5 | 0.88 | 0.75 | 0.73 | 0 | 0 | 0.00001 |
| 2 | 51 | 3 | 0.5 | 0.48 | 0.89 | 0.77 | 0.71 | 0 | 0 | 0 |
| 2 | 51 | 5 | 0.44 | 0.49 | 0.85 | 0.72 | 0.69 | 0 | 0 | 0 |
| 2 | 51 | 10 | 0.45 | 0.45 | 0.84 | 0.71 | 0.67 | 0 | 0 | 0 |
| 3 | 21 | 1 | 0.55 | 0.54 | 0.88 | 0.79 | 0.76 | 0 | 0 | 0 |
| 3 | 21 | 2 | 0.52 | 0.48 | 0.9 | 0.73 | 0.72 | 0 | 0 | 0.00001 |
| 3 | 21 | 3 | 0.47 | 0.44 | 0.88 | 0.75 | 0.74 | 0 | 0 | 0 |
| 3 | 21 | 5 | 0.45 | 0.44 | 0.88 | 0.72 | 0.69 | 0 | 0 | 0 |
| 3 | 21 | 10 | 0.42 | 0.41 | 0.88 | 0.72 | 0.69 | 0 | 0 | 0 |
| 4 | 11 | 1 | 0.57 | 0.58 | 0.84 | 0.76 | 0.75 | 0 | 0.00001 | 0.00002 |
| 4 | 11 | 2 | 0.48 | 0.48 | 0.92 | 0.78 | 0.75 | 0 | 0 | 0 |
| 4 | 11 | 3 | 0.46 | 0.47 | 0.89 | 0.76 | 0.69 | 0 | 0 | 0 |
| 4 | 11 | 5 | 0.44 | 0.44 | 0.88 | 0.72 | 0.69 | 0 | 0 | 0 |
| 4 | 11 | 10 | 0.43 | 0.43 | 0.87 | 0.69 | 0.67 | 0 | 0 | 0 |
| 5 | 9 | 1 | 0.54 | 0.55 | 0.93 | 0.8 | 0.74 | 0 | 0 | 0.00001 |
| 5 | 9 | 2 | 0.49 | 0.48 | 0.93 | 0.8 | 0.74 | 0 | 0 | 0 |
| 5 | 9 | 3 | 0.45 | 0.44 | 0.89 | 0.72 | 0.69 | 0 | 0 | 0 |
| 5 | 9 | 5 | 0.41 | 0.44 | 0.86 | 0.74 | 0.69 | 0 | 0 | 0 |
| 5 | 9 | 10 | 0.4 | 0.44 | 0.92 | 0.71 | 0.67 | 0 | 0 | 0 |
| 6 | 9 | 1 | 0.48 | 0.49 | 0.9 | 0.78 | 0.75 | 0 | 0 | 0 |
| 6 | 9 | 2 | 0.41 | 0.47 | 0.9 | 0.79 | 0.77 | 0 | 0 | 0 |
| 6 | 9 | 3 | 0.41 | 0.4 | 0.87 | 0.73 | 0.69 | 0 | 0 | 0 |
| 6 | 9 | 5 | 0.38 | 0.38 | 0.9 | 0.75 | 0.72 | 0 | 0 | 0 |
| 6 | 9 | 10 | 0.37 | 0.36 | 0.85 | 0.71 | 0.66 | 0 | 0 | 0 |

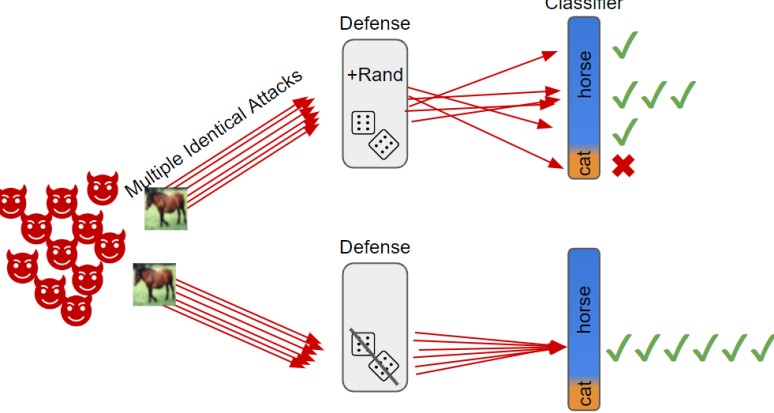

Figure 6: Compared to deterministic defenses, random defenses are more vulnerable to an attacker (or a large group of users) that can submit the same input many times.

file, email spam filter against a mass spam campaign, many users submitting the same query to a ChatGPT-like service).

### D.1 EMPIRICAL COMPARISON OF DEPLOYMENT RANDOMNESS

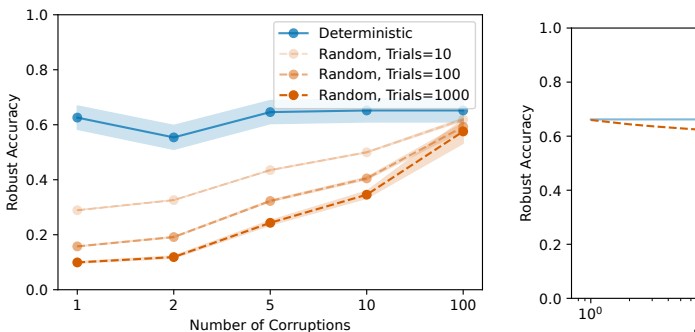
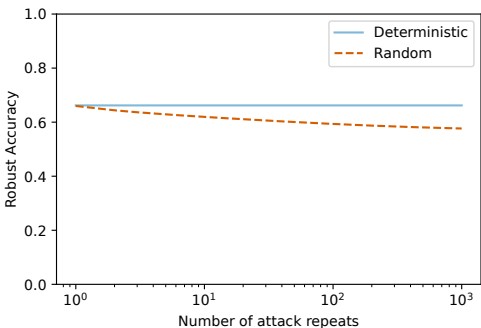

(a) Robust accuracy of randomized smoothing vs deterministic smoothing at varying corruptions. Shaded regions indicate 95% confidence intervals.

(b) Robust accuracy of randomized smoothing vs deterministic smoothing at 100 corruptions.

Figure 7: As an attacker repeatedly attempts to evade correct classification, the robust accuracy of a randomized classifier decreases below its deterministic analog.

As described in section 2.2, Randomized Smoothing makes a prediction by executing a Monte Carlo estimation of the distribution of classes around the input by inferring the class of several copies of the input, each with a separate randomly sampled corruption.

To make this process deterministic, we fix the random seed before every prediction and use the same set of corruptions for every inference. We refer to this deterministic version of inference as *Deterministic Smoothing*.

We compare the robustness of Deterministic Smoothing and Randomized Smoothing against an attacker willing to simply repeat their inference many times to get at least one misclassification. Mirroring the original Randomized Smoothing work (Cohen et al., 2019), we use CIFAR-10 (Krizhevsky, 2009) test datapoints for our experiments. This comparison was completed by computing the classification result of 100,000 Gaussian-corrupted copies of each test datapoint using the *base classifier* (i.e., the classifier being defended). As in prior work, this base classifier has been trained with Gaussian-sampled noise-corruptions up to an $L_\infty$ radius of 0.5. (Cohen et al., 2019) on the CIFAR-10 training set.

In both Randomized Smoothing and Deterministic Smoothing, the number of inferences $n$ aggregated for a single prediction can be adjusted to control for performance and accuracy requirements. We vary this parameter to observe its affect on robustness. With a setting as low as $n = 1$ (which is far below the $n = 100$ used in the original work (Cohen et al., 2019)), Randomized Smoothing's prediction is the class inferred from one noise-corrupted inference through the base classifier. This setting predictably leads to a higher variance of predictions given the same input, while higher values of $n$ leads to lower variance predictions. In contrast, Deterministic Smoothing with $n = 1$ (or $n$ equal to any positive integer), uses the same corruption(s) to corrupt the image for every prediction, so there is no variance in predictions on the same input.

To calculate predictions from Randomized Smoothing using different values of $n$, we shuffle and group the 100,000 base classifier inferences into groups of size $n$, then calculate each of those groups' mode to get the most commonly inferred class within each group. From these $\lfloor \frac{100000}{n} \rfloor$ predictions, we then calculate the probability of misclassification for each individual datapoint. In contrast, Deterministic Smoothing's prediction does not change given the same datapoint and value of $n$. For this reason, we only calculate one prediction for each (test datapoint, $n$) pair. Therefore, as Deterministic Smoothing either gets its one prediction right or wrong, its probability of correct classification for each test datapoint is calculated as 0% or 100%.

Given these probabilities for each datapoint, figure 7a compares the robust accuracy of both Randomized Smoothing and Deterministic Smoothing over the test dataset using different numbers of corruptions $n$ as we increase the number of trials an attacker is allowed. In all cases, Randomized

Smoothing is found to be less robust than Deterministic Smoothing, even at the published defense's chosen value of $n = 100$ (Cohen et al., 2019).

As mentioned in section D, the probability a classifier (assuming independent trials) $f$ predicts the class of a given datapoint correctly $N$ times in a row is $p^N$ where $p$ is the probability any given prediction on that datapoint is correct. However, as just mentioned, deterministic classifiers correctly classify a datapoint in either $0\%$ or $100\%$ of trials, which means $p^N = p$ for deterministic classifiers. This is why Deterministic Smoothing accuracy appears as a horizontal line in figure 7b, but Randomized Smoothing's robust accuracy deteriorates as we increase the number of trials.

