# OpenReview forum: "Subspace Grid-sweep: ML Defense Evaluation via Constrained Brute-force Search"
_ICLR.cc/2024/Conference — Submitted to ICLR 2024_

### Official Review · Reviewer_2Hxt · 2023-10-30

**Soundness:** 3 good
**Presentation:** 3 good
**Contribution:** 3 good
**Rating:** 5
**Confidence:** 3

**Summary:**

This paper introduces a new defense evaluation tool called Subspace Grid-sweep. This tool uses deterministic inference to assess adversarial robustness more effectively, revealing vulnerabilities in a previously published defense. The paper also suggests that randomness may not be necessary for defense robustness.

**Strengths:**

1. The paper is well-written, and I comprehended it easily.
2. The motivation of this paper seems good and well-founded, as initially, researchers believed that randomness enhanced defense robustness. However, with random defenses being overcome, it remains unclear whether randomness is beneficial or merely complicates defense evaluation.
3. The proposal Subspace Grid-sweep is simple and intuitive.

**Weaknesses:**

1. The only difference between deterministic defense and random defense is merely the choice between randomly selecting once and randomly selecting multiple times. In fact, random selection even just once still qualifies as random defense. This may not necessarily imply that defense does not require robustness.
2. The idea of subspace grid sweep lacks novelty.

**Questions:**

1.Why do the results in Figure 2 and Figure 7(a) show differences?

---

> ### Author Response · Authors · 2023-11-16
>
> Thanks to the reviewer for considering our work well-written, well-motivated, and easy to understand. We respond to and ask clarification on the noted reviewer concerns and questions below, and hope this makes the strengths and usefulness of our paper more apparent. Please let us know any other information we can provide.
>
> > The only difference between deterministic defense and random defense is merely the choice between randomly selecting once and randomly selecting multiple times. In fact, random selection even just once still qualifies as random defense. This may not necessarily imply that defense does not require robustness.
>
>
> We categorize a defense as deterministic if it always gives the same output given the same input. If we use the same random seed for each inference in a randomized defense that depends on that seed, then we achieve this effect.
>
> Our reasoning behind this categorization is that one of the difficulties with evaluating randomized defenses is that for each inference, including gradient computation, you get a random variable for the gradients and the output that could be different given the same input. This makes it difficult to do fine-grained optimization when you have randomness in the mix. Also, black-box attacks are hindered, along with other issues discussed in Section 6. Modifying a defense to instead always have the same output given the same input effectively removes these issues, and makes the defense deterministic for our purposes.
>
> We are happy to include some discussion in the paper on why we refer to using the same seed for every inference as making the defense deterministic, and why it suits our purposes.
>
> As for the semantics on if using the same random seed for each inference counts as “removing randomness”, we are also happy to include a discussion on the different definitions of having randomness in a defense.
>
> > The idea of subspace grid sweep lacks novelty.
>
> Could you point us to work that shows ours lacks novelty?
>
> > 1.Why do the results in Figure 2 and Figure 7(a) show differences?
>
> Figure 2 does show that deterministic smoothing is less robust than randomized smoothing when they use lower numbers of corruptions, but converges to the same robustness as the number of corruptions they use increase (implying randomness is not required for empirical robustness for randomized smoothing as discussed in Section 4.1, and it is more important instead to use a sufficient number of corruptions).
>
> Figure 7a is part of Section D in the appendix, which, separately, describes a new threat model where the attacker repeatedly tries the same input to defeat a defense. Figure 7a shows that a random defense will become less robust than a deterministic defense given this threat model (hence the gap).

---

> > ### Comment · Reviewer_2Hxt · 2023-11-22
> >
> > Thanks for the rebuttal. I will keep my rating.

---

### Official Review · Reviewer_LfCf · 2023-10-31

**Soundness:** 2 fair
**Presentation:** 2 fair
**Contribution:** 3 good
**Rating:** 5
**Confidence:** 4

**Summary:**

The paper proposes the Subspace Grid-sweep method. It argues that using a grid for evaluating robustness can provide additional perspective on existing robustness evaluation methods. Also the authors argue that randomness introduced in many popular defences could be redundant and fixed random seed could be used instead.

**Strengths:**

1. Originality. Using structured search for robustness evaluation was used before, in particular, for adversarial patches. See e. g. Derandomized Smoothing [1]. However, using it for Lp bounded perturbations seems to be novel. A look at making randomized attacks deterministic is also interesting.

2. Quality. The paper proposed a simple and interesting idea. The authors discuss the limitations of their method.

3. Clarity. The main idea is easy to grasp.

4. Significance. Evaluating model robustness is important for safety-critical applications. Providing additional perspective on robusness evaluation methods makes them more interpretable.

[1] Alexander Levine and Soheil Feizi. (De)Randomized Smoothing for Certifiable Defense against Patch Attacks. In NeurIPS 2020.

**Weaknesses:**

1. Clarity/Quality. In the main paper and Appendix I haven’t found the explanation on how exactly Auto-Attack was restricted to the continious K-dimensional space? Given that K is extremely small (1-6), this could drastically affect the overall Auto-Attack performance. Without understanding that it is hard to interpret the results e. g. in Table 1.

2. Quality. In Section 4.1 the Authors evaluate Randomized Smoothing with standard PGD. Although they claim to use up to 10 000 steps, it still doesn’t seem like a suitable evaluation method. A-PGD or AutoAttack would be better.

3. Quality. If I understood correctly, all the evaluations were performed on small-resolution datasets (CIFAR-10, SVHN, MNIST). However, Randomized Smoothing is able to provide provable defence e. g. for models on ImageNet. Given the exponential nature of the method, it is hard to understand whether it would scale to datasets of higher resolution.

4. In the limitations the authors admit that fixing random seed in e. g. Randomized smoothing makes it vulnerable for an attacker that knows it. I assume that with the knowledge of Subspace Grid-sweep’s grid  a model could be trained that seems robust when evaluated in the predefined grid points but contains adversarial regions in-between. That could result in backdoor-attacks on seemingly robust models. Thus the method can provide false sense of security.

**Questions:**

1. What is the role of parameter σ in GridSweepPerturbationsL2(K, B, M, σ)? (Section 3.1)

2. How exactly was the orthonormal basis chosen (Section 3.1)?

---

> ### Author Response · Authors · 2023-11-16
> **Response from Authors 1/2**
>
> We would like to thank the reviewer for considering our work as interesting, easy to understand, and important. Also, we thank the reviewer for listing out their questions and concerns. Below, we respond to each of them providing clarification and paper edit proposals. We hope this alleviates the noted concerns and confusion and allows the strengths and usefulness of our work to be clearer.
>
> > Clarity/Quality. In the main paper and Appendix I haven’t found the explanation on how exactly Auto-Attack was restricted to the continious K-dimensional space? Given that K is extremely small (1-6), this could drastically affect the overall Auto-Attack performance. Without understanding that it is hard to interpret the results e. g. in Table 1.
>
> Running AutoAttack in this K-dimensional basis was accomplished by simply adding the K x D matrix (discussed in Sec 3.1) that defines the subspace to the forward function right before the input (where there are K dimensions in the subspace and D is the shape of the original input). This is essentially the same as adding an encoder to the input where AutoAttack modifies the basis weights (and receives backpropagated loss for the basis weights) for the K orthonormal basis vectors that define the subspace instead of the individual pixels of the input. In addition to this straightforward implementation, we know AutoAttack works fine in the subspaces by looking at the results in Tables 1 and 2 that show AutoAttack’s A-PGD finding nearly every vulnerable datapoint (99.8%) that Subspace Grid-sweep found on the undefended (base) classifiers.
>
> The code implementing this forward function for AutoAttack is method `forward_with_basis_vectors(...)` in lines 464-482 in `code/core.py` in the supplementary material.
>
> > Quality. In Section 4.1 the Authors evaluate Randomized Smoothing with standard PGD. Although they claim to use up to 10 000 steps, it still doesn’t seem like a suitable evaluation method. A-PGD or AutoAttack would be better.
>
> We believed this evaluation seems the most appropriate as it is the same attack used (and recommended) by the original authors of Randomized Smoothing (Cohen et al., 2019), and so we wanted to show that Deterministic Smoothing would have been evaluated to have the same robustness as Randomized Smoothing in the original paper. Using a different attack to compare the robustness would be beside the point of the comparison.
>
> However, we actually did use AutoAttack’s A-PGD on Deterministic Smoothing as reported in Section 5 to compare with Subspace Grid-sweep’s ability to find vulnerable datapoints in lower-dimensional subspaces and found no indication that Deterministic Smoothing is especially more vulnerable to A-PGD than PGD. Therefore, as we suspect Randomized Smoothing would also not be especially more vulnerable to A-PGD than PGD, our findings do not change if we consider A-PGD instead of PGD in Section 4.1. These results can be highlighted if the reviewer prefers.
>
>
> >Quality. If I understood correctly, all the evaluations were performed on small-resolution datasets (CIFAR-10, SVHN, MNIST). However, Randomized Smoothing is able to provide provable defence e. g. for models on ImageNet. Given the exponential nature of the method, it is hard to understand whether it would scale to datasets of higher resolution.
>
> As the purpose of Subspace Grid-sweep is to be comparison point to standard evaluation attacks (to see if they are finding vulnerable points that Subspace Grid-sweep finds), then our reported experiments showing issues with defenses using smaller-resolution datasets are still valid, as a defense that is obscuring gradients in a lower-resolution dataset is sufficient cause for concern.
>
> While we have no reason to believe using a larger dataset would not work (as we would still only search a subspace of less than 10 dimensions in a higher-resolution input space), we do not understand any benefit of doing so when we show that using lower-resolution (and much easier to experiment with) datasets is sufficient to show the issue in modern defenses.

---

> ### Author Response · Authors · 2023-11-16
> **Response from Authors 2/2**
>
> > In the limitations the authors admit that fixing random seed in e. g. Randomized smoothing makes it vulnerable for an attacker that knows it. I assume that with the knowledge of Subspace Grid-sweep’s grid a model could be trained that seems robust when evaluated in the predefined grid points but contains adversarial regions in-between. That could result in backdoor-attacks on seemingly robust models. Thus the method can provide false sense of security.
>
> To clarify, we posed as future work to explicitly explore if the attacker somehow has an edge by knowing the random seed randomized smoothing is fixed to. However, our experimental results in Sections 4 and 5 and Figure 2 instead found empirical evidence that the attacker does not have an edge, at least with current attacks used for evaluation.
>
> Also, as later shown in Section 5 using Subspace Grid-sweep in Table 2 where (Smoothing) Defended has a higher Robustness against Grid-sweep than (Smoothing) Base, we show that the attacker may have a hard time using the knowledge of the random seed because there are simply smaller adversarial regions (i.e., many fewer adversarial examples) that can be found when using randomized smoothing, regardless of whether it is random or deterministic.
>
> In response to the posed threat model on if knowledge of Subspace Grid-sweep’s subspaces and grid sizes could allow an adversary to modify their defense to make Subspace Grid-sweep incorrectly deem a defense robust, we point out the following characteristics of Subspace Grid-sweep that should alleviate concerns.
> * Subspace Grid-sweep is not designed to be a standalone *attack*. The purpose of Subspace Grid-sweep is to check whether benchmark attacks (such as AutoAttack’s A-PGD) are finding vulnerable datapoints that should be found, so AutoAttack’s A-PGD is also executed against the defense being evaluated. In the case of an adversary specifically evading Subspace Grid-sweep, they would also need to ensure that AutoAttack’s A-PGD also cannot find these vulnerable regions placed in-between grid points.
> * Additionally, the subspaces for Subspace Grid-sweep are chosen at random (via randomly sampled orthonormal basis vectors), and many subspaces and grids are sampled for an evaluation (shown in Appendix A), so the attacker would need to know, and figure out how to avoid, all the different subspaces and grids of all future evaluators that may want to test the defense.
> * Finally, we also do a random search in the subspaces (results in Appendix A), that would not adhere to the known grid points, and this could also find any hidden adversarial regions places in-between grid points
>
> > What is the role of parameter σ in GridSweepPerturbationsL2(K, B, M, σ)? (Section 3.1)
>
> We apologize for the confusion. $\sigma$ represents the $L_2$ constraint distance that the generated perturbations must adhere to. We are happy to change this to $\epsilon$ to match how we refer to the constraint in the text.
>
> > How exactly was the orthonormal basis chosen (Section 3.1)?
>
> The orthonormal basis is chosen randomly by using a seed defined by the dimensionality of the subspace (K) and the number of bins (B) requested. This is to ensure that the same subspace is used to compare Defended and Undefended (base) classifiers in the several different subspaces we evaluate with (enumerated in Appendix A).
>
> First, we randomly sample a $K \times D $ matrix (after setting the seed), transpose it, use `scipy.linalg.orth` to create orthogonal columns, transpose it again, divide each row by its $L_2$ norm, then exhaustively verify all K rows are orthonormal.
>
> For more detail, the code calculating the orthonormal basis vectors that define the subspaces we search in is in lines 134-146 in `code/basis_experiments.py` in the supplementary material.

---

> ### Comment · Reviewer_LfCf · 2023-11-21
> **Reviewer's response**
>
> Thanks for your response.
>
> Do I understand correctly that the one of the points of your paper and, in particular, of the Table 1 in Section 3.2 is to show that existing attacks (e. g. A-PGD in Table 1) may provide false sense of security by not finding all vulnerable datapoints? In that case I am not sure that you are really showing that since you are significantly restricting A-PGD by appending that KxD matrix to it and, essentially, obtain another artificial attack to which you compare in your low-dimensional space. Could you clarify how your results show the flaws of original A-PGD in the original input space?
>
> Concerning the high-dimensional space, I agree that your approach with up to 10 dimensions can still be applied to it. The question is whether your results will look the same, because by downsampling a 224x224 image to a 10-dimensional space you lose much more information than when you downsample a 32x32 image to a 10 dimensional space. You state that your approach was not meant as a standalone attack but I still wonder whether it maintains it’s useful properties when applied in combination with attacks on datasets that are closer to the real-world problems than e. g. CIFAR10.

---

> ### Author Response · Authors · 2023-11-23
>
> We appreciate your detailed questions! We hope our responses below help clarify our approach.
>
> > Do I understand correctly that the one of the points of your paper and, in particular, of the Table 1 in Section 3.2 is to show that existing attacks (e. g. A-PGD in Table 1) may provide false sense of security by not finding all vulnerable datapoints? In that case I am not sure that you are really showing that since you are significantly restricting A-PGD by appending that KxD matrix to it and, essentially, obtain another artificial attack to which you compare in your low-dimensional space. Could you clarify how your results show the flaws of original A-PGD in the original input space?
>
> This understanding is mostly correct, except that the flaw we find is in the defense being evaluated, not A-PGD. We would extend your first sentence to say that Table 1 shows that AutoAttack’s A-PGD is providing a false sense of security **when used to attack k-Winners-Take-All (k-WTA)**. This symptom of A-PGD failing to find vulnerable points is the result of the k-WTA defense obfuscating gradients (as found in other work (Tramèr et al., 2020)), and not from a flaw in A-PGD.
>
> We can see A-PGD works fine in the lower-dimension subspace when k-WTA is not applied (the first rows of Table 1a and 1b) and can find all vulnerable datapoints. In fact, in all cases where a defense was not used (in Tables 1 and 2), A-PGD successfully found 99.8% of all vulnerable datapoints, indicating that its performance was not hindered at all by searching in a smaller subspace. Intuitively, this makes sense since we are actually giving A-PGD an easier job (only needing to search a 1-6 dimensional space for an adversarial perturbation instead of a 1000+ dimensional space).
>
> In short, the only difference between the first and second rows of Table 1a and 1b is that in row 1, the kWTA defense is not being used, and A-PGD works fine and can find all vulnerable datapoints that grid-sweep can. In contrast, in row 2, where the only difference is that kWTA is being applied, A-PGD is not able to find vulnerable datapoints that grid-sweep can find. The finding of these results is that kWTA is hindering A-PGD from finding existing vulnerable datapoints as the only difference between these two rows is the absence or presence of k-WTA.
>
>
> > Concerning the high-dimensional space, I agree that your approach with up to 10 dimensions can still be applied to it. The question is whether your results will look the same, because by downsampling a 224x224 image to a 10-dimensional space your lose much more information than when you downsample a 32x32 image to a 10 dimensional space. You state that your approach was not meant as a standalone attack but I still wonder whether it maintains it’s useful properties when applied in combination with attacks on datasets that are closer to the real-world problems than e. g. CIFAR10.
>
> To clarify, we are not downsampling the input image to a lower dimension, we are simply restricting the dimensionality of the space of perturbations allowed to be applied to the image by both grid-sweep and A-PGD. The classifier and defense still gets the normal input image, but with a constrained space of adversarial perturbations added to it.
>
> Our answer to the previous question also provides some answer to your concern about us not using ImageNet. To figure out whether a defense is hindering A-PGD, we simply need to compare A-PGD’s performance to Grid-sweep when the defense is absent to when the defense is present. The dataset used only matters in that it does not change between comparisons. We used CIFAR10 since it was used by the original defenses in their evaluation and experiments could be completed much faster (since each model inference is faster) than if we used Imagenet. Switching this comparison from using CIFAR10 to Imagenet would essentially just make the experiments more compute intensive, but could give us no more of a signal on whether a defense is obfuscating gradients than what we have already learned on CIFAR10. That being said, if the reviewer still feels an ImageNet experiment is necessary, we will include one on Randomized Smoothing in Section 5 in our revision. Unfortunately, these experiments will take a couple of weeks to run, so we cannot include them immediately.

---

> > ### Comment · Reviewer_LfCf · 2023-12-04
> >
> > I thank the reviewers for very detailed response. I am increasing my score.

---

### Official Review · Reviewer_EJoi · 2023-11-01

**Soundness:** 2 fair
**Presentation:** 3 good
**Contribution:** 2 fair
**Rating:** 5
**Confidence:** 4

**Summary:**

The authors propose a method to evaluate the effectiveness of defenses in enhancing an image classifier's robustness to adversarial attacks. At the core of many defenses is inherent randomness, which causes difficulty in evaluating whether the defense is truly effective or is merely obfuscating gradients. The authors study whether randomized defenses retain their robustness if they are made deterministic and find that deterministic defenses can be just as robust against white-box attackers as randomized defenses.

**Strengths:**

* The paper studies an interesting problem: How do we correctly evaluate defenses against adversarial attacks?

* The paper has nice Figures and is well-written
* The paper features a detailed Appendix with more information

**Weaknesses:**

**Implication.** On a high level, I do not understand what the authors advocate regarding using randomness in evaluating a defense. While they show that deterministic defenses are, at best, as good as their randomized counterparts, I do not understand in what sense the randomness limits the evaluation of a defense. In other words, if an attack succeeds against a deterministic variant of a defense, what implications does that have on the robustness of the randomized version?

**Subspace Projection.** What is the advantage of using a grid in a subspace over simply randomly sampling noisy perturbations? Also, function $\texttt{GridSweepPerturbationsL2}$ is unclear to me. What is $\sigma$ (I assume this should be $\epsilon$)? For g=0 and B=6, the new g assignment is -1, but the comment is misleading because it says the points are normalized between [0,1].

**Randomized Smoothing.** The insights gained from Section 4.1 is not conclusive to me. The authors state the following.

> This indicates that the empirical robustness of Randomized Smoothing does not come from
the randomness of the noise, but from the self-ensembling effect of aggregating multiple inferences
within proximity of the original point.

Figure 2 shows that a white-box attacker achieves the same success rate regardless of whether the noised perturbations are being resampled during the optimization procedure. However, the inability of an attacker to find an adversarial example does not imply that the randomness does not help. If the attacker knows the corruptions used by the defender, more successful attacks are conceivable.

**Minor.**

* Table 1 is confusing to me. Why is the robustness higher for the undefended model? What is the difference between the first column and the second and third?

**Questions:**

* Why is it meaningful to study the robustness of deterministic defenses?

* What is the advantage of iterating over a grid in a subspace rather than over randomly sampled noisy perturbations?

---

> ### Author Response · Authors · 2023-11-16
> **Response part 1/2**
>
> We would like to thank the reviewer for considering our work interesting, well-written, and detailed, which we are happy to see is a common note in the reviews. Below, we provide responses that we hope alleviate the reviewer’s concerns. Please let us know if there are any other questions we can answer.
>
> > I do not understand in what sense the randomness limits the evaluation of a defense
>
> A more detailed discussion of the problems with randomness in evaluation is in Section 6. A brief summary of these issues is that including randomness in a defense prevents an evaluator from using several useful tools (e.g., black-box attacks, checking whether the underlying gradient is smooth, Subspace Grid-sweep). This extra difficulty in evaluating some random defenses, as shown in Section 4, may not be necessary for the defense to operate and can obscure otherwise easier to find issues (as discussed in Section 5). A recent example of this is concurrent work that had to create a specialized attack to show that DiffPure is masking/obscuring gradients, and is not as robust as originally reported last year [A]. However, our analysis in Section 5 shows this finding in a simpler way with the observation that Subspace Grid-sweep is able to find adversarial examples that AutoAttack’s A-PGD cannot.
>
> > In other words, if an attack succeeds against a deterministic variant of a defense, what implications does that have on the robustness of the randomized version?
> > Why is it meaningful to study the robustness of deterministic defenses?
>
>
> This is a good question. Whether a deterministic variant failing indicates a flaw in the original random variant likely depends on the specific defense being evaluated. If we find that the deterministic variant is showing obfuscated gradients, then adding randomness likely does not solve the obfuscated gradients. However, if a deterministic defense is much less robust to APGD than the random version, then this could mean different things based on what the details of the defense are, but it at least means that the unpredictability of the randomness to the attacker is important for this defense to work, as with only that component removed the defense is degraded. Overall, it gives more information to the evaluator.
>
> > What is the advantage of using a grid in a subspace over simply randomly sampling noisy perturbations?
> > What is the advantage of iterating over a grid in a subspace rather than over randomly sampled noisy perturbations?
>
> As we understand it, there are two parts to this question: (1) Why use a grid search instead of random sampling? And (2) Why search in a lower-dimensional subspace? We’ll give our reasoning on the first question and then the second.
>
> The advantage of using a grid is that it allows us to get a sort of *ground-truth* for what adversarial regions exist that should be able to be found by gradient-guided attacks usually used for evaluation (such as AutoAttack’s A-PGD). As a grid search checks every point at evenly distanced intervals within a space, a grid search ensures that no regions larger than the interval between grid points will be missed. This is in contrast to random sampling that, from bad luck, could miss a large adversarial region in a space by not choosing a point there. However, exhaustively doing a grid-search in a large dimensional space is prohibitive.
>
> Hence, the answer to question 2. The reason we do grid-search in a lower-dimensional subspace is to make it feasible. By also constraining gradient-guided attacks to the same subspace, we have a sort of ground-truth (for regions of size above the interval between grid points) for what datapoints an attack should find an adversarial example for.
>
> On top of this, we also do random search in our lower-dimensional subspace as an extra measure to find adversarial regions that could be hiding between grid points (results in Appendix A).
>
>
> > Also, function GridSweepPerturbationsL2 is unclear to me. What is $\sigma$ (I assume this should be epsilon)? For g=0 and B=6, the new g assignment is -1, but the comment is misleading because it says the points are normalized between [0,1].
>
> Thank you for catching this typo! You are correct, the comment should say “scale grid to [-1,1] and origin center”. Also, you are correct that $\sigma$ represents the $L_2$ constraint distance. We are happy to change this to $\epsilon$ to match how we refer to the constraint in the text.

---

> ### Author Response · Authors · 2023-11-16
> **Response part 2/2**
>
> > However, the inability of an attacker to find an adversarial example does not imply that the randomness does not help. If the attacker knows the corruptions used by the defender, more successful attacks are conceivable.
>
> As discussed in Section 6, we posed as future work to explicitly explore if the attacker somehow has an edge.
>
> However, for the purposes of this paper in showing whether currently used evaluation attacks empirically evaluate robustness well, this edge does not show up given benchmark attacks. Some explanation of why this edge does not show is discussed in Section 5 using Subspace Grid-sweep in Table 2 where (Smoothing) Defended has a higher Robustness against Grid-sweep than (Smoothing) Base, which implies there are simply smaller adversarial regions (i.e., many fewer adversarial examples) that can be found when using randomized smoothing, regardless of whether it is random or deterministic.
>
> > Table 1 is confusing to me. Why is the robustness higher for the undefended model? What is the difference between the first column and the second and third?
>
> You have read this table correctly, and this shows the main result from Section 3, in that kWTA-defended models are actually less robust than the original undefended versions. Our Subspace Grid-sweep shows this issue without catering an attack specifically to kWTA as previous works had to do (Tramèr et al., 2020).
>
> The first column shows robustness against Grid-sweep, as defined in algorithm 2 (1 - (number of vulnerable datapoints / number of datapoints)). The second
> and third columns show the number of vulnerable datapoints Grid-sweep found and how many of those AutotAttack’s A-PGD found. The idea behind these columns is to make it easily visible when AutoAttack’s A-PGD is not finding many of the vulnerable datapoints that Grid-sweep is finding, which would indicate the defense is hindering AutoAttack’s A-PGD from finding existing adversarial regions/examples.
>
> [A] Zhao, Zhengyu, et al. "Towards Good Practices in Evaluating Transfer Adversarial Attacks." arXiv preprint arXiv:2211.09565 (2022).

---

> > ### Comment · Reviewer_EJoi · 2023-11-18
> > **Revised paper**
> >
> > Thank you for your responses. They make sense to me! Could you please provide a revised paper with all the promised changes (ideally highlighted in some color)?

---

> > > ### Author Response · Authors · 2023-11-19
> > >
> > > Certainly! We are glad to hear our responses helped. We have posted a new draft with the changes to Algorithm 1 highlighted in yellow in response to your feedback. Did you mean for us to also implement the changes we suggested in response to other reviewers? We're happy to do that if you think those other changes are helpful, but we were waiting on responses from the other reviewers to confirm that those changes would address their concerns.

---

> > > > ### Comment · Reviewer_EJoi · 2023-11-22
> > > >
> > > > Yes, I meant all revisions. I would like to see everything that changes in the paper. Thank you

---

> > > > > ### Author Response · Authors · 2023-11-23
> > > > >
> > > > > We have posted a new draft with changes highlighted that we proposed in our rebuttal to other reviewers. We appreciate your time in considering our rebuttal and revision!

---

### Official Review · Reviewer_YZ44 · 2023-11-02

**Soundness:** 3 good
**Presentation:** 3 good
**Contribution:** 3 good
**Rating:** 6
**Confidence:** 3

**Summary:**

Recent studies have demonstrated that many defenses, which appear to defend leading-edge attacks at first, may eventually fail against an adaptive attacks. Defenses can seem effective when adversarial examples aren't discovered due to issues like gradient masking, limited computing resources, or poor initial conditions. In this paper, the authors propose a step forward in improving defense assessment by developing a new tool called Subspace Grid-sweep. This tool uses deterministic inference to simplify the process of assessing adversarial robustness. By applying Subspace Grid-sweep, the authors illustrate that a defense method previously thought to be effective—and later proven ineffective—could have been identified as vulnerable without conducting an exhaustive adaptive attack. Furthermore, to extend the Subspace Grid-sweep’s utility to defenses that incorporate randomness, the authors demonstrate methods for creating deterministic versions of these random defenses that maintain comparable empirical performance. Consequently, their findings suggest that randomness might not be essential for the robustness of such defenses.

**Strengths:**

1. This paper unveiled an interesting finding that some randomness-based defenses do not rely on their randomness for the improved robustness.
2. The authors leveraged some interesting methods to transform the randomness-based defenses to deterministic ones to show their robustness.
3. The paper is well-organized and clearly presented.

**Weaknesses:**

1. The grid-sweep-based attacks seem to be very straightforward, and I think if the hidden space is large, the search space for this attack could be still huge.
2. The experiments of diffPure are not clear to me. I believe that diffPure still needs some randomness to be successful otherwise it would be no different from a plain denoiser network. Why is the robust accuracy still as high as around 65% in Figure 4?

**Questions:**

Please refer to the weakness section

---

> ### Author Response · Authors · 2023-11-16
>
> Thanks so much for your review! We are happy to hear you found our work interesting, well-organized, and easy to understand. We respond to the weaknesses you point out below, and hope this allows the paper strengths and usefulness to be increasingly apparent.
>
> > The grid-sweep-based attacks seem to be very straightforward, and I think if the hidden space is large, the search space for this attack could be still huge.
>
> We agree that if the subspace dimensionality is too large (i.e., more than 20), then it would stop making sense to exhaustively search it with subspace grid-sweep. However, the underlying point of the evaluation is not to prove that a defense is robust, but to see if the current attacks being used to evaluate a defense are good enough. For that, we found that using much smaller subspaces (1-6 dimensions) did pretty well, as discussed in Sections 3 and 5, with more detail given in the Appendix. For this reason, we do not think the search space needs to be large to check whether gradient-guided attacks are being hindered by a defense.
>
> > The experiments of diffPure are not clear to me. I believe that diffPure still needs some randomness to be successful otherwise it would be no different from a plain denoiser network. Why is the robust accuracy still as high as around 65% in Figure 4?
>
> This is a good question! Firstly, we are pretty certain that this is the correct result, since we use the exact same code as the original DiffPure paper, except that we fix the random seed to be the same before each inference. We suspect that there is some gradient masking occurring that does not allow AutoAttack to find adversarial examples, which is also supported by our Subspace Grid-sweep results with DiffSmall in Section 5. Fortunately, some recent concurrent work came to the same conclusion by spending a lot more effort specifically examining DiffPure, and found compelling evidence that DiffPure has obfuscated gradients that prevent AutoAttack from finding valid adversarial examples [A]. Our results using Subspace Grid-sweep also discover these issues, and they do so in a more straightforward and general way.
>
> [A] Zhao, Zhengyu, et al. "Towards Good Practices in Evaluating Transfer Adversarial Attacks." arXiv preprint arXiv:2211.09565 (2022).

---

> > ### Comment · Reviewer_YZ44 · 2023-11-20
> > **Response**
> >
> > Thanks for the rebuttal. I will keep my positive rating at 6.

---

### Meta-Review · Area_Chair_sMn5 · 2023-12-08

**Metareview:**

The author suggest to do an exhaustive grid search on a lower-dimensional subspace to find adversarial examples. Simultaneously they run an existing gradient-based attack on this subspace (in the paper mainly APGD) and check if all vulnerable points found by the grid search are also found by APGD. If this is not case then this is a sign that the gradient-based attack is not working well.

The reviewers like the simplicity of the method and consider the paper to be well-written. The scores of the reviewers are borderline.

In my point of view there are several issues with this paper:
- the reported problems in evaluation are known: the fact that APGD (APGD on which loss? could not find this information in the paper) does not get robust accuracy to zero for the k-winners take all-defense is already reported in the original paper of Croce et al. introducing APGD (they show that the black-box Square attack is more efficient in this case). The finding that DiffPure is much less robust than reported has been recently discussed in
  Lee, Kim, Robust evaluation of diffusion-based adversarial purification, CVPR 2023
 Other diffusion-based defenses were discussed earlier in: Croce et al, Evaluating the Adversarial Robustness of Adaptive Test-time Defenses, ICML 2022

- Regarding the randomized defenses, using a deterministic version of the defense is up to my knowledge a standard technique. The authors should then also show that the found adversarial examples transfer to the randomized defense again.

- The authors miss the work of
  Zimmermann et al, Increasing confidence in adversarial robustness evaluations, NeurIPS 2022
  who also have the goal of testing adversarial attacks by implanting known adversarial vulnerabilities.

In total I think this paper includes a nice and simple idea, but the authors would have to show more evaluations on existing defenses in order to validate that their technique is a reliable tool to detect problems in adversarial robustness evaluation. Also the trade-off of the computational cost of the exhaustive search and how reliably the technique works is a bit explored in the appendix but this should be investigated in more depth.
Also the request of one reviewer to test this on ImageNet is important as a technique only working on low-resolution datasets is likely to be of limited interest in the future.

**Justification For Why Not Higher Score:**

see above

**Justification For Why Not Lower Score:**

N/A

---

### Decision · Program_Chairs · 2024-01-16

Reject